# The Role of Mosquito Hemocytes in Viral Infections

**DOI:** 10.3390/v14102088

**Published:** 2022-09-20

**Authors:** Victor Cardoso-Jaime, Chinmay Vijay Tikhe, Shengzhang Dong, George Dimopoulos

**Affiliations:** W. Harry Feinstone Department of Molecular Microbiology and Immunology, Johns Hopkins Bloomberg School of Public Health, Baltimore, MD 21205, USA

**Keywords:** hemocytes, virus, immunity, mosquito, antiviral, virus dissemination

## Abstract

Insect hemocytes are the only immune cells that can mount a humoral and cellular immune response. Despite the critical involvement of hemocytes in immune responses against bacteria, fungi, and parasites in mosquitoes, our understanding of their antiviral potential is still limited. It has been shown that hemocytes express humoral factors such as TEP1, PPO, and certain antimicrobial peptides that are known to restrict viral infections. Insect hemocytes also harbor the major immune pathways, such as JAK/STAT, TOLL, IMD, and RNAi, which are critical for the control of viral infection. Recent research has indicated a role for hemocytes in the regulation of viral infection through RNA interference and autophagy; however, the specific mechanism by which this regulation occurs remains uncharacterized. Conversely, some studies have suggested that hemocytes act as agonists of arboviral infection because they lack basal lamina and circulate throughout the whole mosquito, likely facilitating viral dissemination to other tissues such as salivary glands. In addition, hemocytes produce arbovirus agonist factors such as lectins, which enhance viral infection. Here, we summarize our current understanding of hemocytes’ involvement in viral infections.

## 1. Introduction

Since the discovery that mosquitoes can transmit infectious diseases, hundreds of pathogens, such as parasites and viruses, have been identified as etiologic agents of important human illnesses [1]. *Aedes* mosquitoes are the vectors for many arboviral diseases, such as dengue, Zika, Chikungunya, yellow fever, Mayaro virus, and others, all of which have a significant impact on human health [2,3]. For many years, the primary vectors, *Aedes aegypti* and *Aedes albopictus*, were restricted geographically to parts of Africa and Asia. However, both species have great adaptability to many environments, and globalization and global warming have facilitated the invasion and establishment of these species on every continent except Antarctica [4,5,6,7,8,9]. As a consequence, dengue prevalence has increased about 30-fold during the last 50 years, and it is now considered the most important mosquito-borne viral disease. An estimated 400 million dengue cases annually occur globally, and more than half of the world’s population is at risk of infection [10,11].

In the last decades, significant efforts have been focused on eradicating mosquito-borne diseases; however, diseases such as malaria remains the cause of more than 200 million cases and about half a million deaths each year. The success in reducing malaria cases since 2000 is largely attributable to vector control [2,12]: insecticides and mosquito repellents have been the main tools used to successfully limit vector populations and pathogen transmission, but unfortunately mosquitoes have developed resistance to these insecticides, which has led to a re-emergence of vector-borne disease [13]. As an alternative strategy to overcome these obstacles, the use of genetically engineered mosquitoes to suppress the vector population or reduce vector competence is currently being evaluated [14].

In the field, only about 1% of mosquitoes actually contract arboviral infections, mainly thanks to the mosquitoes’ immune responses, which are a critical factor in limiting viral infection [15,16]. The mosquito immune response involves cellular and humoral responses that eliminate pathogens such as bacteria, fungi, parasites, and viruses. The humoral response includes the release of antimicrobial peptides (AMPs), prophenoloxidase (PPO), and opsonins such as thioester-containing proteins (TEPs). Most of these antimicrobial molecules are produced by epithelial cells, mainly in the fat bodies; the exceptions are PPO and the TEPs, which are exclusively produced by hemocytes [17]. In addition, hemocytes are responsible for the mosquito’s cellular immune response, which involves phagocytosis, encapsulation, and nodulation [18]. In the case of viruses, the antiviral immune responses involve apoptosis of infected cells and virus RNA targeting by the RNA interference (RNAi) pathway [19,20,21]. Other than the RNAi pathway, most of the known immune processes are modulated by four major signaling pathways: TOLL, IMD, JAK/STAT, and JNK [21]. During the past two decades, the molecular mechanisms of various immune pathways have been uncovered, specifically by studying the immune responses in the midgut and the salivary glands, which are the critical organs for the development and transmission of human pathogens in the mosquito [22,23]. Nevertheless, it is well known that hemocytes are exclusive producers of critical molecules, such as TEP1, PPO, and sP22D, which limit *Plasmodium* development in the mosquito [24,25,26,27]. It is very likely that hemocytes are involved in the antiviral immune response. However, this aspect of hemocytes is poorly understood. In the present review, we discuss the existing research that supports an antiviral role for hemocytes.

## 2. The Role of Hemocytes in Antiviral Immunity

Phagocytosis is one of the most evolutionarily conserved mechanisms of immune defense. In mammals, the cells specialized for phagocytosis are known as macrophages, whereas in invertebrates, including insects, they are known as hemocytes [28].

As in mammals, whose phagocytic cells are comprised of subpopulations with specialized functions (e.g., monocytes, macrophages, neutrophils, and basophils), insect hemocytes can also be subdivided into cell types with specialized functions [29]. Most mosquitoes have three main hemocyte subpopulations that differ in their morphology: granulocytes, oenocytoids, and prohemocytes. The main function of granulocytes is phagocytosis, but they are also involved in the production of PPO, opsonins, and AMPs; oenocytoids are associated with the production of PPO; and prohemocytes have characteristics of stem cells [18,26,30].

Hemocytes are the only immune cell type that can move throughout the whole insect body. They usually circulate through the hemolymph (circulating hemocytes), but they can also attach to specific tissues and organs, such as infected tissues or cells and sites of injury (sessile hemocytes) [18,31]. For example, in *Anopheles gambiae*, infections induce an interaction between the immune and circulatory systems, whereby circulating hemocytes became sessile hemocytes attached to the heart [32,33]. These sessile hemocytes easily phagocytize pathogens circulating in the hemolymph [34]. Similarly, when *Plasmodium* ookinetes invade the mosquito midgut epithelium, they cause cell damage; as a consequence, hemocytes are recruited specifically to the infected cells, where they release vesicles for complement activation against *Plasmodium* [35].

Despite the various mechanisms and functions of hemocytes that have been discovered in multiple insect species, including mosquitoes, most studies have focused on the hemocytes’ immune responses against bacteria and parasites. In mosquitoes, hemocytes have been proposed to act as either agonists or antagonists of viral infection, but there is not enough current evidence in mosquitoes to definitively determine their role. However, since all insects have hemocytes with similar subpopulations and functions [29,36], it is possible to speculate about their shared mechanisms of immune response against viruses.

Hemocytes are the only mosquito cells without basal lamina, and its absence facilitates the entry and release of viruses within these host cells [37,38,39]. Several reports have shown that dengue virus serotype 2 (DENV2), Sindbis virus (SINV), and West Nile virus (WNV) have tropism for hemocytes of *Ae. aegypti* [40,41,42]. Similarly, SINV has tropism for the hemocytes of *Ae. albopictus*, *Aedes triseriatus*, and *Culex pipiens* [43]; also, o´nyong-nyong virus (ONNV) shows tropism for the hemocytes of *An. gambiae* [44]. Since hemocytes can become infected and support virus replication, Engelhard et al. (see [38]) have suggested that hemocytes are the amplification center for viruses in the hemolymph, but not a determinant for virus dissemination, because all the other tissues are associated with a basal lamina that is non-permeable to viruses [45,46]. Free viruses circulating in the hemolymph are likely to first infect the tracheas that surround tissues, and then infect their cells [38,39].

Several studies have demonstrated the various ways in which viruses can be disseminated from the midgut to other organs. One of these is that the infected midgut cells first promote virus replication, and then the tracheal system in the midgut becomes infected and supports the replication of the virus and its release into the hemolymph. The virus circulating in the hemolymph infects and replicates in tissues such as the fat body, hemocytes, and salivary glands, and these tissues support virus replication as well [37,38,40,47,48,49].

Meanwhile, in the midgut, the mosquito’s blood meal triggers the synthesis of enzymes involved in blood digestion and induces tissue distension and apoptosis; all of these factors alter the midgut epithelium’s structure, enabling the virus to pass through the basal lamina and be released into the hemolymph to infect secondary tissues [46,50,51,52,53]. Many pathogens (e.g., *Plasmodium* sporozoites) use the flow of the hemolymph as transport to reach various tissues, such as the salivary glands [54]; similarly, viruses can use the hemolymph for transport to various tissues. Many reports have shown that mosquitoes become infected through the injection of viruses into the hemocoel [38,43,47,51,53], supporting the hypothesis that the viruses retain their integrity and infectivity in the hemolymph long enough to infect the tissues. In fact, hemocytes support the replication of a virus in the same way it occurs in other cells; however, there is no evidence that only hemocytes facilitate the transport of virus through the whole body of the mosquito.

In contrast, several reports have shown that phagocytic cells appear to control viral infections. In mice, depletion of macrophages results in higher dengue virus titers [55]. In *Drosophila*, it has been shown that hemocytes play a critical role in controlling systemic viral infection (the mechanism is discussed below) [56,57,58]. In the mosquito *Ae. aegypti*, it has recently been reported that inhibition of phagocytosis results in higher systemic dissemination and replication of DENV2 and Zika virus (ZIKV), suggesting that phagocytosis is required for arbovirus control in the mosquito [59]. Despite these pieces of evidence indicating that hemocytes play a role in controlling viral infection, they are also known to produce immune factors that act as agonists, suggesting that, in some way, hemocytes may have two opposing functions. In the next section, we discuss the possible agonistic and antagonistic activities of hemocytes in viral infections.

## 3. Antiviral Immune Pathway in Hemocytes

Insects harbor four major conserved immune pathways: the small interfering RNA (siRNA) pathway, the JAK-STAT pathway, the TOLL pathway, and the IMD pathway [21]. Of these four, the siRNA pathway is considered the primary antiviral pathway in insects, but the other three pathways have also been shown to play important roles in antiviral defense. Multiple studies have shown that these pathways are important for insect hemocyte biology; however, evidence linking these immune pathways to the hemocytes’ antiviral capacity is still lacking.

### 3.1. The siRNA Pathway

The siRNA pathway is composed of three major proteins: Dicer-2 (Dcr2), double-stranded RNA binding protein (dsRBP) R2D2, and Argonaute-2 (Ago2). In *Bombix mori*, Dcr2 is highly expressed in the hemocytes, and expression levels have been further shown to increase in these cells after nucleopolyhedroviral (BmNPV) infection [60]. Knocking down Dcr2 also increases BmNPV viral titers. However, a single-cell RNA-sequencing study of BmNPV-infected hemocytes in *B. mori* has shown suppression of the RNAi pathway by the virus [61]. These studies highlight the role of the RNAi pathway in *B. mori* hemocytes; however, its direct involvement in the antiviral defense remains unclear [62].

In *Ae. aegypti,* overexpression of the key genes Dcr2 and R2D2 produces a broad-spectrum defense against multiple arboviruses, including DENV2, ZIKV, and Chikungunya virus (CHIKV) [63]. Ago2 is present in the soluble fraction of the hemolymph, and its expression has been shown to be upregulated upon blood feeding [64]. However, it was unclear in this study whether the Ago2 present in the soluble fraction of the hemolymph was secreted or somehow processed by the hemocytes. Another antiviral protein, P400, is highly expressed in *Ae. aegypti* hemocytes, and this protein regulates Ago2 gene expression; it has been suggested that antiviral activity in *Ae. aegypti* hemocytes can be regulated by P400 through the alteration of the expression of Ago2 [65].

The systemic response to dsRNA molecules in insects is highly variable. Multiple studies have demonstrated that the insect species, the route of dsRNA entry, and the tissue involved are three key factors that determine the intensity of the systemic RNAi response [66,67,68]. Unlike nematodes and plants, insects lack specialized dsRNA transporters. Insect hemocytes have long been speculated to play a role in the systemic RNAi response. A detailed role for the RNAi pathway in *Drosophila* hemocytes has been demonstrated in a comprehensive study by Tassetto and colleagues [56]. They reported that *Drosophila* lacking hemocytes and those with a reduced expression of Ago2 in their hemocytes show increased SINV titers in the whole body. Despite this increase in the virus titer throughout the body, the hemocytes were not infected with SINV. These findings highlighted an indirect role for this RNAi pathway protein in mediating the systemic antiviral response and also demonstrated that hemocytes readily take up viral RNA and then use reverse transcriptase to convert it into viral DNA (vDNA); this vDNA serves as a template for producing secondary viral siRNAs. These viral siRNAs are secreted from the hemocytes in the form of exosomes whose contents provoke a systemic antiviral response throughout the body. The hemocytes were also found to secrete exosomes long after the infection, providing suggestive evidence of an adaptive antiviral immune response. Thus, this study has revealed a role for the RNAi pathway in *Drosophila* hemocytes in manifesting systemic antiviral responses [56].

Even though the hemocytes have been shown to be responsible for systemic RNAi responses in *Drosophila*, the same mechanism has not been substantiated as yet in other insects. Systemic RNAi responses have not yet been definitively demonstrated in many insects, and the mechanism by which hemocytes take up viral RNA has not been established. In the *Ae. aegypti* mosquito, the Toll 6 receptor has been implicated in dsRNA sensing and uptake that activates the Toll pathway [69]; this study was carried out in the *Ae. aegypti* phagocytic cell line Aag2, suggesting that phagocytic hemocytes may use a similar mechanism to recognize and take up viral RNAs. Clearly, the role of the RNAi pathway and hemocytes in non-drosophilid insects is an interesting area for future research.

### 3.2. The Toll Pathway

The Toll pathway plays a key role in hemocyte biology. In *Drosophila*, it is essential for larval hematopoiesis [70]. Flies with an activated Toll pathway have an increased number of hemocytes in the hemolymph and show a melanotic tumor phenotype [71,72]. The Toll pathway also plays an important role against viruses *in Drosophila*, honeybees, the small brown planthopper *Laodelphax striatellus,* and *Aedes* mosquitoes [73,74,75,76,77]. In *An. gambiae*, it has been reported that Toll over-activation induces hemocyte differentiation, enhancing the immune response against *Plasmodium* [78]. Even if there is no evidence as yet of the involvement of the TOLL pathway in the mosquito hemocytes’ antiviral response, the mechanisms are likely shared between mosquito and other insect species.

In *Drosophila*, the hemocytes at the site of tissue injury show an elevated expression of the Toll pathway. There is also an increased production of ROS at the injury site, which prevents further infections [79]. The ROS and Toll pathways have been implicated in *Wolbachia*-mediated blocking of DENV in *Aedes* mosquitoes [80]. Whether the ROS–Toll pathway connection and its antiviral effect are mediated via hemocytes remains unstudied. In *Aedes* mosquitoes and *Drosophila*, phenoloxidase genes are regulated by the Toll pathway [81]. Other than the melanization response, Toll-like receptors expressed in the hemocytes may act as Toll pathway activators by sensing viral dsRNAs, leading to an antiviral response [69].

The antiviral role of the Toll pathway is not well established in *B. mori*. However, a bacterial pathogen recognition receptor, the scavenger receptor C (SR-C) protein, is highly expressed in *B. mori* hemocytes [82] and leads to the activation of the Toll pathway. Whether this activation also results in an antiviral effect mediated via hemocytes is unclear and would be an interesting question to pursue.

### 3.3. The JAK-STAT Pathway

Like the Toll pathway, the JAK-STAT pathway also plays an important role in hemocyte biology. It also acts as a key antiviral immune pathway in insects [83,84]. The JAK-STAT pathway is essential for hemocyte maturation and differentiation in *Drosophila* [71]. In mosquitoes, STAT is expressed in several immunocompetent tissues, including the fat body and hemocytes [85]. A cytokine-like protein, unpaired 3 (upd3), secreted by hemocytes, is an activator of the JAK/STAT pathway. Upd3 has been implicated in responses to injury, insulin sensitivity, lifespan determination, and defense against parasitoid wasps in *Drosophila*. The gene encoding this protein is also upregulated in periosteal hemocytes of *An. gambiae* after a bacterial infection [86]. Even though this pathway plays important roles in both hemocyte biology and general antiviral responses, studies connecting these two areas are lacking.

### 3.4. The IMD Pathway

The IMD pathway has been shown to be involved in the antiviral immune response in multiple insect species [73,87,88]. In *Drosophila*, IMD pathway mutants infected with cricket paralysis virus (CrPV) show an increased viral titer [87]. Flies with injected beads that block the phagocytic activity of the hemocytes also show increased CrPV titers [58]. However, it is unclear whether activation of the IMD pathway is mediated in or by the hemocytes. IMD pathway activation in hemocytes leads to increased resistance to *S. aureus* infection, but whether IMD activation is also associated with antiviral defense is not known [89]. Interestingly, in another study, injection of Epstein–Barr virus (EBV) was found to result in increased hemocyte proliferation and an induction of the IMD pathway. Silencing the IMD pathway also resulted in a decrease in hemocyte proliferation after EBV DNA injection [90]. This study highlighted a connection between *Drosophila* hemocytes and the IMD pathway-mediated response to viral DNA. Other than these few studies, however, evidence connecting the IMD pathway to hemocyte-mediated antiviral responses is sparse.

### 3.5. The JNK Pathway

The c-Jun N-terminal kinase (JNK) pathway is highly conserved in eukaryotes. In insects, the JNK pathway regulates several physiological processes affecting immunity and homeostasis, such as metabolism, cell proliferation, apoptosis, and antimicrobial response [91]. In *Ae. aegypti*, the JNK pathway restricts viral infection in the salivary glands by upregulation of complement factor TEP20 and apoptosis activator Dronc [92]. Interestingly, the JNK pathway regulates the expression of anti-plasmodial effectors TEP1 and FBN9 in hemocytes of *An. gambiae* [93]. In addition, the IMD and JNK pathways regulate periosteal aggregation, phagocytosis, and melanization associated with hemocytes, most probably by the regulation of TEP expression [86]. Recently, it was reported that in *An. albimanus*, pericardial cells (also called nephrocytes that are cells that surround the periosteal regions) express immune genes, such as those encoding lysozymes and IMD, suggesting that they participate in the mosquito’s immune response [94,95]. In *Drosophila*, pericardial cells regulate hemocyte accumulation in the heart in response to JNK pathway activation [96,97], suggesting a complex interaction between hemocytes and other tissues. In *Ae. aegypti,* hemocytes are attached to the midgut during viral infection [59]; however, the mechanism governing this interaction and its effects on infection are still unknown. More studies are needed to determine whether this regulation occurs similarly in the heart and whether it has an impact on viral infection.

The majority of studies on hemocyte immune pathways have been performed in *Drosophila*. However, most of these studies have looked at isolated immune responses against a particular type of pathogen. Thus, at the global level, the role of individual immune pathways and their crosstalk in hemocyte-mediated antiviral responses remain largely unknown.

## 4. Humoral Factors Produced by Hemocytes Regulate Viral Infections

### 4.1. Prophenoloxidase

Hemocytes produce several soluble molecules, such as AMPs, enzymes, and opsonins, to eliminate pathogens circulating in the hemolymph. PPO is produced exclusively by hemocytes, and along with its role in pathogen defense, it is also involved in coagulation, cuticle hardening, and pigmentation [98,99,100]. The PPO system is one of the most important mechanisms for the elimination of bacteria, fungi, and *Plasmodium*. Interestingly, PPO upregulation and activation have been reported in *Ae. aegypti*, *Armigeres subalbatus*, and *Lymantria dispar* infected with Semliki forest virus, SINV, and baculovirus Lymantria dispar multiple nucleocapsid nucleopolyhedrovirus (LdMNPV), respectively. In addition, knockdown or inhibition of PPO increases the viral load and mortality in insects, suggesting that PPO is involved in the antiviral immune response [33,34,35]. It has been suggested that PPO suppresses viral infection by killing infected cells (damaged by cell lysis) through melanization [101], or through recognition of the glycoprotein of the viral envelope by lectins that activate the PPO cascade [102]. However, the mechanism remains unknown, and it may be specific for a particular virus type and insect species.

### 4.2. Antimicrobial Peptides (AMPs)

The AMPs are the most conserved humoral effectors of immune systems. They are present in all living organisms, but in insects they display one of the broadest degrees of diversity and highest levels of abundance [103,104]. In insects, the AMPs are produced by various tissues, with fat bodies and hemocytes being the most common sources [105]. The AMPs exhibit activity against almost all kinds of pathogens, including viruses [104,106]. Interestingly, recent reports have shown that DENV infection in *Ae. aegypti* and various mosquito cell lines induces overexpression of AMPs such as defensins, cecropins, gambicin, diptericin, and attacin [69,107,108]. In addition, it has been reported that a knockdown of cecropin and defensin genes increases the viral load in *Ae. aegypti*, suggesting an antiviral role for these AMPs [107]. Even though there is no evidence thus far to suggest that mosquito hemocytes express AMPs as an antiviral mechanism of defense, they express most of the antimicrobial peptides, including cecropins and defensins [86,109,110,111,112]. In *B. mori*, baculovirus infection induces upregulation of cecropins in hemocytes, which probably have antiviral functions [61]. It would not be a surprise for mosquito hemocytes to play a similar role; however, more studies are needed to address this hypothesis.

### 4.3. Pattern Recognition Receptors (Opsonins)

The innate immune response starts with the recognition of non-self molecules by pattern recognition receptors (PRRs). Insects, which lack a classic adaptive immunity, have instead developed a broad repertoire of PRRs that recognize common structures in pathogens known as pathogen-associated molecular patterns (PAMPs). These PAMPs include the lipopolysaccharide of Gram-negative bacteria, peptidoglycan of Gram-positive bacteria, β-1,3-glucan in fungi, and the dsRNA or ssDNA of viruses. After PAMPs have been recognized, PRRs act as opsonins, or they activate and regulate the immune response through immune signaling pathways that trigger the production of humoral factors, phagocytosis, encapsulation, and nodulation (reviewed in [113,114]). Curiously, mosquito hemocytes express the most common PRRs as well as some exclusive PRRs that are involved in phagocytosis, melanization, or nodulation [112]. It is also interesting that hemocytes have been reported to express PRRs that participate in viral elimination, but they also express other PRRs that play an opposite role, facilitating viral infections.

In insects, one of the most important PRRs is the family of thioester-containing proteins (TEPs), which are essential for antibacterial defense [115,116,117]. In mosquitoes, TEPs are a critical factor in the elimination of *Plasmodium* during the early stages of infection [24,35,118,119]. Hemocytes haves been suggested as the main producers of TEPs [24,120]. Interestingly, a recent study has shown that knockdown of TEP1 and TEP2 results in higher titers of DENV2 and WNV in *Ae. aegypti* [121,122]. Overexpression of TEP1 also suppresses DENV2 infection [122], suggesting an antagonist role for *Ae. aegypti* TEP1 during DENV infection. Similarly, in *Ae. aegypti,* a macroglobulin complement-related factor (AaMCR) and scavenger receptor-C (AaSR-C) have an antagonist role against DENV1-4 and yellow fever virus (YFV), and they are more highly expressed in hemocytes than in any other tissue [107]. It has been suggested that TEPs provide resistance to flavivirus infection by activating the TOLL, JAK/STAT, and IMD pathways, as well as producing AMPs [92,93,107,121,123]. TEP1, in cooperation with other proteins, can regulate mechanisms such as melanization, AMP expression, and phagocytosis, which can have an impact on viral infection.

Lectins are very important to the insect immune response because they can recognize and bind to carbohydrates in the walls of microorganisms, mainly bacteria. Lectins are involved in the processes of pathogen elimination, including opsonization, PPO activation, encapsulation, nodulation, and agglutination. However, they can also act as agonists with regard to some pathogens (reviewed in [124]). For example, it is well known that in *An. gambiae, P. falciparum* evades the immune response by recruiting CTL4 and CTLM2A [125,126]. In addition, several reports have described lectins as a critical factor in the establishment of arboviral infection in mosquitoes, because they facilitate the virus’s entry into mosquito cells (reviewed in [127]). In *Ae. aegypti,* nine mosquito galactose-specific C-type lectins (mosGCTLs) have been identified that facilitate DENV2 infection, with mosGCLT-3, which is highly expressed on hemocytes, being the most critical [42]. In a similar way, mosGCLT-1 has been implicated in facilitating WNV infection in *Ae. aegypti* and *Culex quinquefasciatus* and is highly expressed in hemocytes and salivary glands as well [41]. Both mosGCLT-1 and mosGCLT-3 bind to protein E of WNV and DENV2, forming a complex lectin/protein E in the hemolymph [41,42]. Then binding of mosGCLT-1 to protein E is recognized by tyrosine phosphatase 1 (mosPTP-1), which is expressed in most tissues, facilitating viral entry into multiple mosquito tissues [41]. Even though there is as yet no supporting evidence, it is probable that mosGCLT-3 and other lectins undergo similar interactions with other receptors.

Lectins can recognize the carbohydrates of viral envelopes, which can lead to the activation of multiple immune response mechanisms, including the PPO cascade and phagocytosis. However, viruses have developed strategies to evade the immune system, in this case using lectins as a receptor for endocytosis and cell invasion. On the other hand, hemocytes express other antagonist PRRs that are conducive to viral elimination, such as TEP. Studies in this area have shown that hemocytes express genes with opposite functions against viruses, and it is probable that hemocytes can play agonist or antagonist roles, depending on many factors that influence the mosquito’s immune response (including the type of virus and insect, the degree of fitness, and the composition of the microbiota). For instance, it is well known that the mosquito microbiota plays an important role in susceptibility to viral infection [128]. In *Ae. aegypti,* mosGCLTs are important contributors to homeostasis of the microbiota, and their expression is regulated by its composition [129]. Thus, an alteration in the mosquito microbiota can alter the expression of mosGCLTs and antagonist factors in the hemocytes, which can cause a switch in the hemocytes from agonist to antagonist (or vice versa) with regard to arboviral infections.

## 5. Cellular Antiviral Immunity in Hemocytes

### 5.1. Apoptosis

Apoptosis is a form of programmed cell death (PCD) that occurs in multicellular organisms and removes old or damaged cells. Apoptosis is considered an important component of multiple processes that contribute to tissue homeostasis, embryonic development, and innate immunity [130]. The regulation of apoptosis in insects is highly conserved and is controlled by initiator and effector caspases and inhibitor of apoptosis (IAP) [131].

In *Ae. aegypti*, apoptosis is established as a component of the antiviral immunity elicited during arbovirus infection [132,133]. Infection with arboviruses, such as DENV, SINV, and CHIKV, has been shown to induce apoptosis in the midguts and salivary glands of the mosquitoes [51,134,135]. In these tissues, arbovirus infection increases the expression of effector caspase genes that stimulate apoptosis to defend the insect against the infection. Apoptosis induced by chemical treatment increases the antiviral defense against arboviral infection and replication, while suppression of apoptosis caused by knocking down caspase genes compromises the mosquitoes’ defense against arboviral infection [51,135]. It is unknown whether apoptosis is activated in hemocytes during arbovirus infection in mosquitoes. However, many apoptotic genes have been shown to be expressed in hemocytes in mosquitoes [110,112], suggesting that the apoptotic pathway is active in mosquito hemocytes and may play a role in restricting the infection of mosquitoes by arboviruses.

### 5.2. Autophagy and Phagocytosis

Autophagy is an evolutionarily conserved process in all eukaryotic cells that serves to degrade unused or old cellular components, such as damaged organelles, defective mitochondria, and misfolded proteins [136]. It plays an essential role in clearing these aggregates on the cytoplasmic surface in order to maintain cellular and tissue homeostasis and promote development and survival. During autophagy, a double phospholipid membrane, called an autophagosome, forms around damaged cellular components, and then fuses with a lysosome to mediate the degradation of the sequestered contents [137]. Autophagy can be induced by starvation or other cellular stresses, such as DNA damage, formation of protein aggregates, or pathogen infection, which increase the number of autophagosomes [138]. Autophagy has been shown to be involved in midgut and salivary gland tissue homeostasis in *Drosophila* [139,140], and vitellogenesis in the fat bodies of *Ae. aegypti* [141], and the midgut cells, silk glands, and fat bodies of silkworms [142,143,144]. Although well studied in midguts and fat bodies, autophagy has been less well studied in hemocytes, except for a study in *Tenebrio molitor*, which indicated that autophagy in hemocytes may play a role in defending this organism against infection with *Listeria monocytogenes* [145].

The antiviral effect of autophagy in mosquitoes has been explored in mosquito cells. In the phagocytic cell line Aag2 [146], infection with DENV2 and CHIKV significantly induced autophagy, but using pharmacological agents to treat cells to induce or inhibit autophagy had no significant effect on arbovirus infection [147]. Another study in the same mosquito cell line, treatment with the autophagy inducer rapamycin was found to activate the autophagic pathway and thus defend against DENV infection [148]. Nevertheless, both studies were conducted with mosquito cell lines, making it difficult to draw a clear conclusion that autophagy has an antiviral effect in vivo in the major arbovirus mosquito vectors. Transcriptome studies using the Mayaro virus (MAYV)–*Anopheles stephensi* system seem to indicate that virus infection downregulates the expression of genes related to autophagy at a late time point [149]. However, in the whitefly *Bemisia tabaci*, infection with a begomovirus, tomato yellow leaf curl virus (TYLCV), activates the autophagy pathway through the regulation of ATG8 production by activating the MAPK signaling cascade, which is modulated by an interaction between phosphatidylethanolamine-binding protein (PEBP) and the coat protein (CP) of TYLCV [150,151].

In *Drosophila* hemocytes, the possible antiviral role of autophagy has been tested for six different viruses, and contradictory results were obtained. Impairment of the autophagy pathway affected infection with SINV, Drosophila C virus (DCV), CrPV, and invertebrate iridescent virus 6 (IIV6). Autophagy played a proviral role in flock house virus (FHV) infection and had a only modest antiviral effect in decreasing infection with vesicular stomatitis virus (VSV), suggesting that autophagy may not represent the major antiviral defense in hemocytes [58]. Thus, it is still unclear whether hemocytes play a role in inhibiting arboviral infection in mosquitoes and other arbovirus vectors, and more studies in this area are clearly needed.

On the other hand, inhibition of phagocytosis in *Drosophila* and *Ae. aegypti* has been found to result in an increase in the systemic titer of arbovirus [57,59]. Interestingly, in *Drosophila*, hemocytes confer protection from virus infection in a manner that is dependent on the phagocytosis of apoptotic cells or uptake of cellular debris [56,57]. In addition, it has been proposed that hemocytes accumulate viral RNA by removing dead infected cells or their debris, and then use the dsRNA to confer viral resistance via the siRNA pathway [56] (discussed in Section 3.1). Therefore, phagocytosis may contribute to the cellular immune responses against viral infection in mosquito hemocytes. However, more effort is needed to clearly test this hypothesis.

## 6. Concluding Remarks

Despite the many advances that have been made in elucidating the molecular basis of mosquito immunology, the mechanisms governing its cellular immune responses against viral infection are still far from clear. Hemocytes are an extremely vital part of the mosquito’s immune system and are involved in both cellular and humoral responses. They exclusively produce several immune factors that are critical for the elimination of human pathogens. Here, we have discussed some of the possible ways that hemocytes participate in the immune response against viruses. Interestingly, the information available suggests that hemocytes could be involved in the elimination of viruses as well as in protecting them from elimination and promoting infection (Figure 1). Hemocyte function can be influenced by external factors, such as the nature of the microbiota, the viral load, fitness, and other factors, which can cause the scale to lean toward an agonistic or antagonistic effect on the viruses. However, it is clear that more research is needed to thoroughly assess the role(s) of hemocytes during viral infections.

For many years, one of the main limitations in hemocyte research was the scarcity of sensitive and accurate tools to examine these cells. Adult mosquitoes have only 2000–5000 hemocytes, so the use of suitable strategies and technologies is particularly critical. Recently, many new tools have been developed for the study of hemocytes, such as intravital staining, which allows the monitoring of hemocytes in vivo [32]. Single-cell RNA sequencing has allowed the identification of different subpopulations in mosquitoes and their specific mechanisms of immune response against pathogens such as *Plasmodium* [109,110,111]. Furthermore, clodronate liposomes have been used to deplete phagocytic hemocytes in *An. gambiae*, *Ae. aegypti*, and *Drosophila* [119,152]. Finally, a promoter of *Drosophila* hemolectin driving mosquito hemocyte-specific expression has been reported [44]. These new tools will help us to address many questions about hemocyte biology, as well as to identify new strategies for vector-borne disease control.

## Figures and Tables

**Figure 1 viruses-14-02088-f001:**
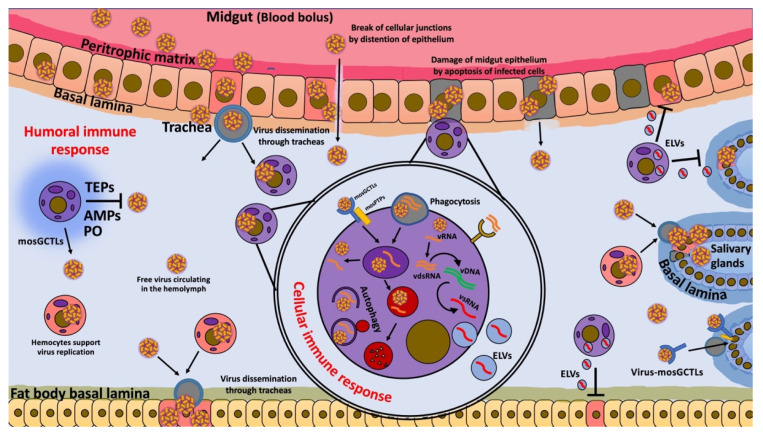
Hemocytes’ immune response during viral infections. Hypothetical model of possible functions of hemocytes during viral infection. Blood meal contains viruses that trigger the production of digestive enzymes and midgut distention, which disrupt cellular junctions and cause apoptosis in midgut epithelium cells (gray square cells). These effects facilitate virus dissemination, allowing infection of cells and trachea via gaps in the cellular junctions and via apoptosis of cells. Viruses released from the midgut epithelium can be freely transported in the hemolymph or by infected hemocytes (circular purple (uninfected) and pink cells (infected)) that are distributed in the flowing hemolymph to secondary tissues, such as fat bodies and salivary glands, which receive the virus through the tracheal system. Hemocytes can facilitate the entry of a virus into cells of various tissues through the production of mosquito galactose-specific C-type lectins (mosGCTLs) and protein tyrosine phosphatases (mosPTPs). However, hemocytes also produce phenoloxidase (PO), antimicrobial peptides (AMPs), and thioester-containing proteins (TEPs), which are involved in virus elimination. In addition, in the later stages of infection, hemocytes take up free viruses by endocytosis or phagocytosis of apoptotic cells, together with cellular debris containing virus and dsRNA; they then develop viral small interfering RNAs (vsRNAs) that are released by exosome-like vesicles (ELVs) to enter infected cells and confer virus resistance through the RNAi pathway. Finally, hemocytes can also eliminate viruses by autophagy.

## Data Availability

Not applicable.

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
