# Peer review of "The Role of Mosquito Hemocytes in Viral Infections"

_viruses, 2022, doi:10.3390/v14102088_

Round 1

Reviewer 1 Report

The authors address an important new topic in mosquito immunity: the roles of hemocytes. The senior author is an authoritative figure in the field of mosquito immunity. The review is comprehensive and addresses the important topics in the field. I found it informative to draw information from other insects than mosquitoes, as little has been done in mosquito hemocytes. The article should be useful to promote studies on deciphering the antiviral functions of mosquito hemocytes. 

I have listed minor comments below that should be addressed before publication.

Introduction:
In the first paragraph, it is stated that 50-100 million dengue cases occur every year. Authors should update their reference to Bhatt et al, 2013 that calculated that about 400 million cases occur yearly.
The second paragraph is mostly about plasmodium vectors. This is confusing as the review is about antiviral immunity. I suggest to significantly reduce this paragraph, even delete it, and blend the information about plasmodium vector in the rest of the text if necessary.
“the main approaches… involve apoptosis and RNA interference”, can the author substantiate this claim that these are the main antiviral pathways, or downgrade the claim.
Among the canonical immune pathways, Chowdhury et al, 2020 recently added the JNK pathway, which is antiviral against CHIKV, DENV and ZIKV. The authors should update their text with this new information.

The role of hemocytes in antiviral immunity
While the section covers the important topics, the rational flow of the paragraphs could be improved by shifting the paragraph order. I suggest to move the fourth paragraph (starting with “Despite the various…”) to the beginning of the section. Move the 7th just before the 6th paragraph.
Also, please substantiate your statements with references to the following statements: “all other tissues contain a basal lamina that is non-permeable to viruses” add ref. “use the tracheal system as a dissemination route to other tissues”.

Antiviral immune pathways in hemocytes
Here as well, please add the JNK pathway as an antiviral pathway, even more so as reference is made to this pathway in relation to hemocytes, later in the text.
This sentence is confusing: “Multiple studies have shown… infected with various viruses.”

The siRNA pathway
I am not aware of a study supporting this statement “hemocyte-like cell line Aag2”. So far, we know that Aag2 can mount an immune response and there are multiple tissues that can do so but we don’t know the tissue origin of the cell line.

4.2. AMPs
“Even though there is no evidence thus far to suggest that… as an antiviral mechanism of defense, these insects express most of the antimicrobial peptides…” Shouldn’t “insects” be replaced with hemocytes?

4.3. opsonins
Sorry to insist but TEP are also induced by JNK and were shown to be antiviral in Chowdhury et al.

Figure 1. While informative, it should be mentioned that a lot of these have not been formally demonstrated in mosquitoes and thus can be considered hypotheses.
Also, the figure would benefit from a legend indicating what drawing represents hemocytes.

Author Response

Comments and Suggestions for Authors

The authors address an important new topic in mosquito immunity: the roles of hemocytes. The senior author is an authoritative figure in the field of mosquito immunity. The review is comprehensive and addresses the important topics in the field. I found it informative to draw information from other insects than mosquitoes, as little has been done in mosquito hemocytes. The article should be useful to promote studies on deciphering the antiviral functions of mosquito hemocytes. 

I have listed minor comments below that should be addressed before publication.

Introduction:
COMMENT 1: In the first paragraph, it is stated that 50-100 million dengue cases occur every year. Authors should update their reference to Bhatt et al, 2013 that calculated that about 400 million cases occur yearly.

RESPONSE: We have added the new data and reference: Lines 34-36, “An estimated 400 million dengue cases annually occur globally, and more than half of the world´s population is at risk of infection [10,11].”.

  1. Bhatt, S.; Gething, P.W.; Brady, O.J.; Messina, J.P.; Farlow, A.W.; Moyes, C.L.; Drake, J.M.; Brownstein, J.S.; Hoen, A.G.; Sankoh, O.; et al. The Global Distribution and Burden of Dengue. Nature 2013, 496, 504–507, doi:10.1038/nature12060.

COMMENT 2: The second paragraph is mostly about plasmodium vectors. This is confusing as the review is about antiviral immunity. I suggest to significantly reduce this paragraph, even delete it, and blend the information about plasmodium vector in the rest of the text if necessary.

RESPONSE: We appreciate this comment, however, we use Anopheles-malaria as an example of vector control -focused disease control strategies, and also because studies on Anopheles hemocytes are much more advanced than those of Aedes. We have rewritten the first sentences accordingly. Lines 36-45, “In the last decades, significant efforts have been focused on eradicating mosquito-borne diseases; however, diseases such as malaria remains the cause of more than 200 million cases and about half a million deaths each year. The success in the reduction of malaria cases since 2000 is largely attributable to vector control [2, 12]: Insecticides and mosquito repellents have been the main tools used to successfully limit vector populations and pathogen transmission, but unfortunately mosquitoes have developed resistance to these insecticides, which has led to a re-emergence of vector-borne disease [13]. As an alternative strategy to overcome these obstacles, the use of genetically engineered mosquitoes to suppress the vector population or reduce vector competence is currently being evaluated [14].”.

COMMENT 3: “the main approaches… involve apoptosis and RNA interference”, can the author substantiate this claim that these are the main antiviral pathways, or downgrade the claim.

RESPONSE: We appreciate this comment, several studies have shown that the RNA interferences pathway and apoptosis represent main defense systems against viruses, as also is suggested in the reviews by Sim, et al., 2014; Tikhe, et al., 2021; (cite 19 and 21) and we added a new reference:

  1. Marques, J.T.; Imler, J.-L. The Diversity of Insect Antiviral Immunity: Insights from Viruses. Current Opinion in Microbiology 2016, 32, 71–76, doi:10.1016/j.mib.2016.05.002.

We also rewrote the sentence, lines 55-59, “In the case of viruses, the antiviral immune responses involve apoptosis of infected cells and virus RNA targeting by the RNA interference (RNAi) pathway [19–21]. Other than the RNAi pathway, most of the known immune processes are modulated by four major signaling pathways: TOLL, IMD, and JAK/STAT and JNK [21].”

COMMENT 4: Among the canonical immune pathways, Chowdhury et al, 2020 recently added the JNK pathway, which is antiviral against CHIKV, DENV and ZIKV. The authors should update their text with this new information.

RESPONSE: We agree with this suggestion and added a new section (3.5 The JNK pathway, Lines 247-270). We move the two last paragraphs of 3.4; The IMD pathway to the new section and we discussed the participation of JNK pathway in the immune response of hemocytes against viruses. We added a new section (3.5 The JNK pathway). We moved the two last paragraphs of the 3.4 section to the 3.5 and we further discussed the participation of the JNK pathway in the immune response of hemocytes against viruses. We also added new references:

  1. Tafesh-Edwards, G.; Eleftherianos, I. JNK Signaling in Drosophila Immunity and Homeostasis. Immunology Letters 2020, 226, 7–11, doi:10.1016/j.imlet.2020.06.017.
  2. Chowdhury, A.; Modahl, C.M.; Tan, S.T.; Xiang, B.W.W.; Missé, D.; Vial, T.; Kini, R.M.; Pompon, J.F. JNK Pathway Restricts DENV2, ZIKV and CHIKV Infection by Activating Complement and Apoptosis in Mosquito Salivary Glands. PLOS PATHOGENS 2020, 16, e1008754., doi:10.1371/journal.ppat.1008754.
  3. Garver, L.S.; de Almeida Oliveira, G.; Barillas-Mury, C. The JNK Pathway Is a Key Mediator of Anopheles Gambiae Antiplasmodial Immunity. PLoS Pathog 2013, 9, e1003622, doi:10.1371/journal.ppat.1003622.

The role of hemocytes in antiviral immunity:
COMMENT 5: While the section covers the important topics, the rational flow of the paragraphs could be improved by shifting the paragraph order. I suggest to move the fourth paragraph (starting with “Despite the various…”) to the beginning of the section. Move the 7th just before the 6th paragraph.

RESPONSE: We appreciate this suggestion; however, we believe that the flow is this section is better served by the order of paragraphs we initially used.

COMMENT 6: Also, please substantiate your statements with references to the following statements: “all other tissues contain a basal lamina that is non-permeable to viruses” add ref. “use the tracheal system as a dissemination route to other tissues”.

RESPONSE: We have added a relevant references to this section:

  • Lines 98-99, “Hemocytes are the only mosquito cells without basal lamina, and its absence facilitates the entry and release of viruses within these host cells [37–39]”.
  • Lines 106-107, “all the other tissues are associated to a basal lamina that is non-permeable to viruses [45,46]”

And we rewrote the last sentence in the same paragraph:

  • Lines 107-108, “ Free viruses circulating in the hemolymph are likely to first infect the tracheas that surround other tissues, and then infect their cells [38,39].”

We also added references:

  1. Keddie, B.A.; Aponte, G.W.; Volkman, L.E. The Pathway of Infection of Autographa Californica Nuclear Polyhedrosis Virus in an Insect Host. Science 1989, 243, 1728–1730, doi:10.1126/science.2648574.

  1. Trudeau, D.; Washburn, J.O.; Volkman, L.E. Central Role of Hemocytes in Autographa Californica M Nucleopolyhedrovirus Pathogenesis in Heliothis Virescens and Helicoverpa Zea. J Virol 2001, 75, 996–1003, doi:10.1128/JVI.75.2.996-1003.2001.

  1. Dong, S.; Balaraman, V.; Kantor, A.M.; Lin, J.; Grant, D.G.; Held, N.L.; Franz, A.W.E. Chikungunya Virus Dissemination from the Midgut of Aedes Aegypti Is Associated with Temporal Basal Lamina Degradation during Bloodmeal Digestion. PLoS Negl Trop Dis 2017, 11, e0005976, doi:10.1371/journal.pntd.0005976.

Antiviral immune pathways in hemocytes
COMMENT 7: Here as well, please add the JNK pathway as an antiviral pathway, even more so as reference is made to this pathway in relation to hemocytes, later in the text.

RESPONSE: We did as outlined in response to comment 4.

COMMENT 8: This sentence is confusing: “Multiple studies have shown… infected with various viruses.”

RESPONSE: We rewrote the sentence. Lines 146-148, “Multiple studies have shown that these pathways are important for insect hemocyte biology, however, evidence linking these immune pathways to the hemocytes’ antiviral capacity is still lacking”.

The siRNA pathway

COMMENT 9: I am not aware of a study supporting this statement “hemocyte-like cell line Aag2”. So far, we know that Aag2 can mount an immune response and there are multiple tissues that can do so but we don’t know the tissue origin of the cell line.

RESPONSE: We change it to “this study was carried out in the Ae. aegypti phagocytic cell line, Aag2, suggesting that phagocytic hemocytes may use a similar mechanism to recognize and take up viral RNAs”; lines 190-191, and “In the phagocytic cell line Aag2 [147], infection with DENV2 and CHIKV significantly induced autophagy, but using pharmacological agents to treat cells to induce or inhibit autophagy had no significant effect on arbovirus infection [148].” lines 404-406.

We have cited an additional study that support the phagocytic activity of the Aag2 cell line:

  1. Barletta, A.B.F.; Silva, M.C.L.N.; Sorgine, M.H.F. Validation of Aedes Aegypti Aag-2 Cells as a Model for Insect Immune Studies. Parasites Vectors 2012, 5, 148, doi:10.1186/1756-3305-5-148.

4.2. AMPs
COMMENT 10: “Even though there is no evidence thus far to suggest that… as an antiviral mechanism of defense, these insects express most of the antimicrobial peptides…” Shouldn’t “insects” be replaced with hemocytes?

RESPONSE: We have corrected this mistake; we rewrote the sentence. Lines 296-399, “Even though there is no evidence thus far to suggest that mosquito hemocytes express AMPs as an antiviral mechanism of defense, they express most of the antimicrobial peptides, including cecropins and defensins”.

4.3. opsonins
COMMENT 11: Sorry to insist but TEP are also induced by JNK and were shown to be antiviral in Chowdhury et al.

RESPONSE: We now mention the JNK pathway as a regulator of TEPs. Lines 327-331, “It has been suggested that TEPs provide resistance to flavivirus infection by activating the TOLL, JAK/STAT and IMD pathways, as well as producing AMPs [92,93,107,122,124]. TEP1, in cooperation with other proteins, can regulate mechanisms such as melanization, AMP expression and phagocytosis that can have an impact on viral infection.”.

COMMENT 12: Figure 1. While informative, it should be mentioned that a lot of these have not been formally demonstrated in mosquitoes and thus can be considered hypotheses. Also, the figure would benefit from a legend indicating what drawing represents hemocytes.

RESPONSE: We now comment that Figure 1 is a hypothetical model, lines 463-465. We say which cells correspond to hemocytes.

Reviewer 2 Report

Thank you for the opportunity to review this manuscript. The topic is original.How the mosquitoes recognize and respond to viral infection is a central question that directly affects their vector competence.In the present review, authors discuss the existing research that supports an antiviral role for hemocytes.

However I have some suggestion to improve this manuscript. In the first place it’s necessary to evidence what it is. Is a narrative review? A systematic review?The research strategy is lacking. It’s necessary to explain it. For good examples of search strategies used in systematic reviews see the Cochrane Database of Systematic Reviews and the Database of Abstracts of Reviews of Effects (DARE). Most published systematic reviews include the full strategies used for each database, often in the appendices. You may be able to combine sections of strategies used in other reviews to form the basis of a good search strategy for your review. In addition, it's necessary to to add a table in which you summarize the evidences.

Author Response

Thank you for the opportunity to review this manuscript. The topic is original.How the mosquitoes recognize and respond to viral infection is a central question that directly affects their vector competence.In the present review, authors discuss the existing research that supports an antiviral role for hemocytes.

However I have some suggestion to improve this manuscript.

COMMENT 1: In the first place it’s necessary to evidence what it is. Is a narrative review? A systematic review?

RESPONSE: It is a narrative review.

COMMENT 2: The research strategy is lacking. It’s necessary to explain it. For good examples of search strategies used in systematic reviews see the Cochrane Database of Systematic Reviews and the Database of Abstracts of Reviews of Effects (DARE). Most published systematic reviews include the full strategies used for each database, often in the appendices. You may be able to combine sections of strategies used in other reviews to form the basis of a good search strategy for your review.

RESPONSE: Thank you for the comment, however, the aim of this review is to discuss different studies related to the role of hemocytes during viral infection”, we didn’t use any database.

COMMENT 3: In addition, it's necessary to to add a table in which you summarize the evidences.

RESPONSE: We appreciate the comment, however, only limited evidence (published studies) shows the direct role of hemocytes in viral infection, and most of the evidence suggest that it is in an indirect way. Figure 1 integrates all evidences in a better (visual) way.

Round 2

Reviewer 2 Report

The revision is not sufficient.